# Reliable Unlearning Harmful Information in LLMs with Metamorphosis Representation Projection

## Abstract

While Large Language Models (LLMs) have demonstrated impressive performance in various domains and tasks, concerns about their safety are becoming increasingly severe. In particular, since models may store unsafe knowledge internally, machine unlearning has emerged as a representative paradigm to ensure model safety. Existing approaches employ various training techniques, such as gradient ascent and negative preference optimization, in attempts to eliminate the influence of undesired data on target models. However, these methods merely suppress the activation of undesired data through parametric training without completely eradicating its informational traces within the model. This fundamental limitation makes it difficult to achieve effective continuous unlearning, rendering these methods vulnerable to relearning attacks. To overcome these challenges, we propose a Metamorphosis Representation Projection (MRP) approach that pioneers the application of irreversible projection properties to machine unlearning. By implementing projective transformations in the hidden state space of specific network layers, our method effectively eliminates harmful information while preserving useful knowledge. Experimental results demonstrate that our approach enables effective continuous unlearning and successfully defends against relearning attacks, achieving state-of-the-art performance in unlearning effectiveness while preserving natural performance. Our code will be available upon publication.

## 1 Introduction

Recently, with the increasing capacity of Large Language Models (LLMs) to learn from vast corpora, growing concerns have emerged regarding their potential to generate private, harmful, or illegal content [30, 29, 17]. In response, regulatory frameworks such as the EU's General Data Protection Regulation (GDPR) [35] and the California Consumer Privacy Act (CCPA) [4] have established the "right to be forgotten", mandating that applications must support the deletion of specific information upon user request. This has spurred significant research interest in machine unlearning [5, 3, 15] to address these challenges.

So far, existing LLM unlearning approaches primarily focus on parameter optimization. Some methods formulate loss functions to achieve model unlearning [39, 9, 42, 18, 21], while others modify model architectures by incorporating unlearning layers or classifiers to enhance unlearning performance [6, 13]. However, recent research [24, 33] reveals that even when unlearning appears to be successful, implicit knowledge may still persist within model parameters, showing the superficiality of existing unlearning methods. In particular, attackers can intentionally recover sensitive information through relearning or jailbreaking attacks. For instance, some adversaries employ adversarial attacks like GCG [45] to enhance the prompts to elicit the model's harmful content generation [26]. Others fine-tune open-source models using benign data similar to the unlearned data [16, 27], effectively

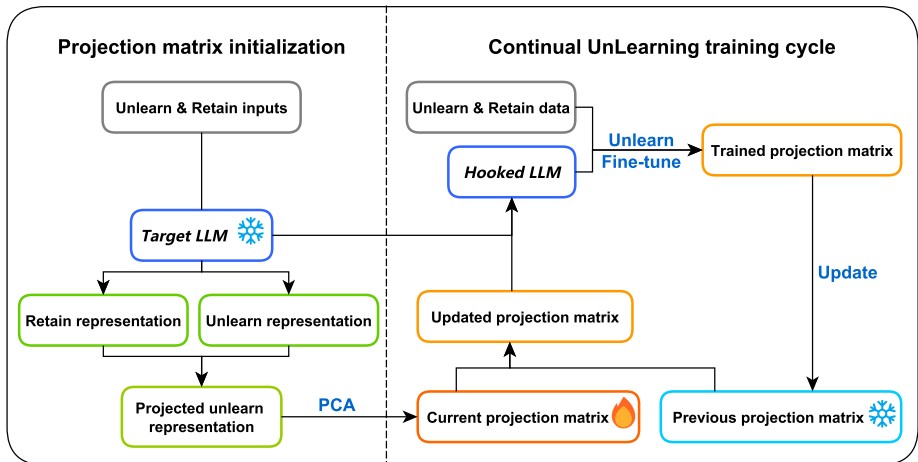

Figure 1: A brief overview of MRP. Our method consists of two key components: Projection Matrix Initialization and Continual Unlearning Training Cycle. For each unlearn data, we first feed both unlearn and retain inputs into the target LLM to extract their hidden state vectors as representations. The unlearning representations are then projected onto the orthogonal complement space of the retention representations. Through PCA, we derive the initial projection matrix, which is combined with the previous projection matrix and integrated into the target LLM to form a hooked LLM. Finally, we perform unlearning fine-tuning on the combined projection matrix using both unlearn and retain data. The trained projection matrix updates the previous projection matrix iteratively, enabling continuous unlearning through successive inputs of unlearn/retain data pairs.

recovering forgotten knowledge. The success of these attacks demonstrates that existing unlearning methods struggle to completely eliminate knowledge-related information from models.

Furthermore, most current unlearning methods handle only a single unlearning request. In practice, however, unlearning requests are sequential [20, 32], as original training data may become inaccessible over time due to expired access rights, privacy concerns, or intellectual property protection [24, 22]. However, existing unlearning methods suffer from catastrophic forgetting in continuous unlearning scenarios [43]. Our experiments confirm this phenomenon and further provide an intuitive explanation: subsequent unlearn tasks may restore certain model parameters, reactivating their forgotten knowledge.

To overcome these challenges, we propose a Metamorphosis Representation Projection (MRP) method that applies projection operations to the model's hidden state vectors. Specifically, we introduce projection matrices after the MLP layers to erase forgotten information from hidden states while ensuring minimal impact on other useful tasks, as illustrated in Figure 1. Experiments demonstrate that our approach achieves strong unlearning effects with minimal parameter updates. Due to the compatibility of projection operations, even if the already projected vector is projected again, the forgotten information will not be restored, making our method particularly effective in continuous unlearning scenarios. Remarkably, by training only 0.1M parameters, our method maintains an unlearning performance score of 0.905 after completing four unlearn tasks, significantly outperforming the best baseline score of 0.785. Moreover, the irreversible nature of projections ensures that once hidden states are modified, unlearned information remains robust against relearning attacks. This property enables our method to maintain an accuracy of 0.383 on unlearn tasks even after five epochs of relearning attacks, while other baselines exhibit minimal accuracy increases to 0.506 on the same tasks.

Our key contributions are summarized as follows:

- We identify limitations in existing unlearning methods and empirically demonstrate that both relearning attacks and continuous unlearning can lead to parameter recovery, reactivating forgotten knowledge.
- We propose a hidden state projection approach, experimentally validating its effectiveness and robustness in removing task-specific information from hidden states.

- Our method achieves superior performance in continuous unlearning scenarios and provides strong defense against relearning attacks, addressing a critical challenge in LLM unlearning.

# 2 Related Work

## 2.1 Machine Unlearning

Machine unlearning was first introduced by [5], with initial applications focusing on compliance with regulatory "right to be forgotten" requirements and eliminating retained knowledge that models should not possess [40, 41]. Significant progress has been made in unlearning across various domains, including image classification [12, 10], text-to-image generation [12, 10], Healthcare AI [31, 19, 1], blockchain [47, 23, 46], and graph neural networks [6, 7, 38].

## 2.2 Language Model Unlearning

Despite advances in unlearning techniques, unlearning in LLMs remains particularly challenging due to their internal complex mechanisms. The enormous parameter space of LLMs makes the precise targeting of unlearning objectives difficult. Existing representative approaches include Gradient Ascent (GA) [39] fine-tunes models by performing gradient ascent on unlearn data while applying gradient descent on retain data, Efficient Unlearning method for LLMs (EUL) [6] introduces an additional unlearning layer after MLP blocks, Negative Preference Optimization (NPO) [42] balances unlearning and retention through a tuning parameter $\beta$, and Representation Misdirection for Unlearning (RMU) [21] perturbs hidden states of unlearned data while preserving original representations for retained data.

However, these methods exhibit two critical limitations: (1) They suffer from catastrophic forgetting when handling sequential unlearning requests, compromising performance on earlier tasks, and (2) They only superficially suppress model outputs without robustly erasing the internal unlearned knowledge representations.

Recent work by [13] proposed the first systematic continuous unlearning framework using Orthogonal Low-Rank Adaptation (O-LoRA) [37] to maintain task independence, combined with an Out-of-Distribution (OOD) detection module to identify data requiring unlearning. While this approach shows improved continuous unlearning performance, it introduces substantial computational overhead for OOD training and suffers from false positives when processing data similar to unlearned examples. Crucially, it still fails to completely eliminate internal knowledge, leaving models vulnerable to relearning attacks.

## 2.3 Model Representation

Our work builds on model representation analysis [44]. In large language models, representations refer to the distributed patterns of neuron activations that encode specific linguistic features or task knowledge across network layers. These representations are typically extracted through forward propagation of input data and analysis of hidden state vectors at targeted layers. Recent studies demonstrate that task-specific representations in LLMs tend to be sparse and low-rank [8]. While some unlearning methods leverage representation manipulation [21] by controlling post-unlearning similarity to original representations, they underutilize these sparsity and low-rank properties.

Our key insight addresses this gap: By extracting orthogonal subspaces of model representations for unlearned data and training projection matrices in these subspaces, we achieve precise elimination of target representations. The inherent sparsity and low-rank structure ensure minimal impact on other tasks' performance while enabling continuous unlearning. Furthermore, the irreversible nature of our projections provides robust defense against relearning attacks, offering a novel solution to fundamental challenges in unlearning.

# 3 Motivation and Preliminaries

## 3.1 Motivation

Most existing model unlearning studies employ fine-tuning approaches that treat unlearning as a learning task, where model parameters are adjusted using both unlearn and retain datasets. This methodology, however, is prone to catastrophic forgetting. To substantiate this claim, we conducted

preliminary experiments using LLama2-7B [36] as our target model and selected three consecutive unlearn tasks from the ScienceQA [25] dataset's natural science module: biology (Task 1), chemistry (Task 2), and physics (Task 3).

As shown in Table 1, when applying the traditional Gradient Ascent (GA) method to Task 1, we observed satisfactory unlearning performance with the QA accuracy dropping from 0.640 to 0.363. However, when proceeding to unlearn Tasks 2 and 3, the unlearning effectiveness on Task 1 deteriorated significantly, with QA accuracy rebounding to 0.411 and eventually 0.447.

To further investigate this phenomenon, we performed two gradient analysis experiments:

- For the model that has unlearned on Task 1, we extracted average layer-wise gradients ($G_{unlearn1}$, $G_{unlearn2}$) when processing Tasks 1 and 2, respectively;
- For the model that has unlearned on Tasks 1 and 2, we extracted gradients ($G'_{unlearn1}$, $G_{unlearn3}$) when processing Tasks 1 and 3.

As shown in Figure 2(a), most layers exhibit negative cosine similarity between $G_{unlearn1}$ and $G_{unlearn2}$, indicating fundamental conflicts between Task 1 and Task 2 unlearn objectives. This explains why unlearn Task 2 partially reverses Task 1's unlearning effects. Figure 2(b) demonstrates even stronger conflicts between $G'_{unlearn1}$ and $G_{unlearn3}$, with lower cosine similarity values. This suggests that unlearning interference becomes more severe as the model undergoes consecutive unlearning operations.

| Unlearn progress | Origin | Task 1 | Task 2 | Task 3 |
|---|---|---|---|---|
| QA Accuracy | 0.640 | 0.363 | 0.411 | 0.447 |

Table 1: Performance of the model on Task 1 during the continuous unlearning process

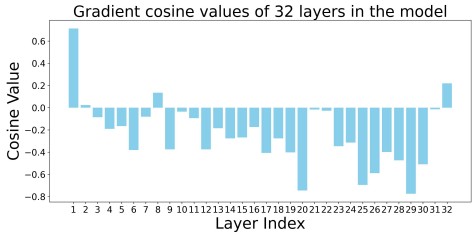
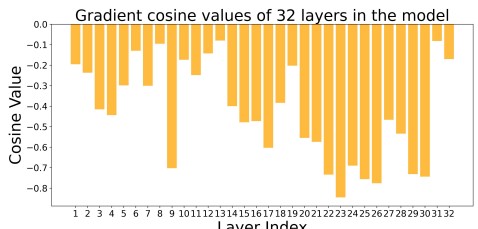

(a) The cosine similarity between $G_{unlearn1}$ and $G_{unlearn2}$

(b) The cosine similarity between $G'_{unlearn1}$ and $G_{unlearn3}$

Figure 2: Layer-wise gradient cosine similarity during unlearning across two tasks

Motivated by observations above, we propose a method capable of completely unlearning Task A while ensuring that the unlearning process for Task B does not interfere with the unlearning of Task A. Instead of overwriting the parameters with a new set of fine-tuned parameters, we conceptualize unlearn Task A as eliminating the information related to Task A from the parameters. To achieve this, we only need to project the model parameters orthogonally to the subspace associated with Task A, effectively removing the information pertaining to Task A from the parameters. Similarly, even if a similar projection operation is performed for Task B, the projected model parameters will remain orthogonal to Task A, thereby minimizing the impact on the unlearning performance for Task A. Overall, this procedure can be viewed as a metamorphosis representation projection (MRP) mechanism.

### 3.2 Preliminaries

**Model architectures**. We formulate the layer-wise components in LLMs as follows. Generally, a (decoder-only) LLM can be formulated as $f^W = g \circ l_T \circ \cdots \circ l_2 \circ l_1 \circ f$, where blocks $\{l_i\}_{i=1}^T$ represent successive layers of the model, consisting of attention modules and MLP modules, $f$ and $g$ denote the encoding and decoding operations respectively, $W$ denotes all parameters of the model.

**Parameter Projection**. To reduce computational resources while demonstrating the efficiency of our approach, we selectively apply projection after certain MLP layers rather than all layers. Let $H$ denote the set of MLP layers that require projection. For layers not in $H$, we consider their projection

transformation as an identity transformation, corresponding to an identity matrix as the projection matrix.

Formally, let $f^{W,P}$ be the modified model after projection. Let Then:

$$f^{W,P} = g \circ P_n \circ l_n \circ P_{n-1} \circ l_{n-1} \circ \cdots \circ P_1 \circ l_1 \circ f \tag{1}$$

where for $h \notin H$, $P_h = I$ (the identity matrix).

After obtaining the task specific representation, we need to find a projection transformation to eliminate the information of the model on the task while not affecting its performance in the retain task, we can achieve this by utilizing the following properties of the projection matrix.

**lemma 3.1.** *Let $P \in \mathbb{R}^{n \times n}$ be a projection matrix. Then there exists an orthogonal matrix $Q \in \mathbb{R}^{k \times n}$ (with $k \leq n$) such that:$P = I - Q^T Q$.*

**lemma 3.2.** *Let $P \in \mathbb{R}^{n \times n}$ be a projection matrix, $Q$ is an orthogonal matrix $Q \in \mathbb{R}^{k \times n}$ such that:$P = I - Q^T Q$, For any $x \in \mathbb{R}^n$, $Px$ is the orthogonal projection of $x$ onto the complement of the row space of $Q$*

The detailed proofs of these two lemmas are provided in the Appendix A.

## 4 Proposed Method: Metamorphosis Representation Projection

### 4.1 Projection Matrix Training

From lemma 3.1, we can train a low-rank matrix $Q$ to achieve the effect of the projection matrix $P = I - Q^T Q$. For the training objective, we employ the standard cross-entropy loss $L(W, D)$ to measure the performance of model $f^W$ with parameters $W$ on dataset $D$. To simultaneously consider both unlearning effectiveness and knowledge retention, we formulate the composite loss function as:

$$\mathcal{L}(W, Q) = -L(W, P, D_{\text{unlearn}}) + \alpha L(W, P, D_{\text{retain}}), \tag{2}$$

where $\alpha > 1$ is a hyperparameter balancing the trade-off (set to 1.2 as default in our experiments), $P = I - Q^T Q$ is the projection matrix, $D_{\text{unlearn}}$ and $D_{\text{retain}}$ denote the unlearn and retain datasets respectively, then the optimization objective becomes:

$$\underset{Q_h (h \in H)}{\arg\min} \left[ -L(W, P_h, D_{\text{unlearn}}) + \alpha L(W, P_h, D_{\text{retain}}) \right] \tag{3}$$

where $P_h = I - Q_h^T Q_h$, $H$ represents the set of selected hidden layers for projection.

### 4.2 Orthogonal Initialization

To ensure effective initialization of $Q$, we leverage lemma 3.2 which suggests that retention task performance is preserved when the row vectors of $Q$ are orthogonal to the retention task's feature space. Our initialization procedure comprises four steps:

1. Select $K$ ($K = 200$) retain task data to extract hidden states $\{h_1^{\text{ret}}, ..., h_K^{\text{ret}}\}$. In order to facilitate the initialization of the projection matrix, we also need to use the QR algorithm [11] calculate their orthogonal basis matrix: $Q_{\text{ret}} = QR(h_1^{\text{ret}}, ..., h_K^{\text{ret}})$

2. Project the unlearn task's hidden states $h_i^{\text{unl}}$ onto the orthogonal complement space spanned by $\{h_1^{\text{ret}}, ..., h_K^{\text{ret}}\}$: $h_{i,p}^{\text{unl}} = (I - Q_{\text{ret}}^T Q_{\text{ret}}) h_i^{\text{unl}}$

3. Use PCA [14] to compute the top-$k$ principal components of the projected hidden states to initialize $Q$'s row vectors:$Q_{\text{init}} = \text{PCA}(h_{1,p}^{\text{unl}}, ..., h_{K,p}^{\text{unl}})$ This initialization guarantees:

   - Orthogonality to retain representation: $Q_{\text{init}} h_i^{\text{ret}} = 0$ for all $i \in [1, ..., K]$,
   - Alignment with unlearn task: $Q_{\text{init}}$ approximates the most significant directions in the unlearn task's residual space.

4. Combine the initialized matrix $Q_{\text{init}}$ with previous matrix $Q_{\text{prev}}$, and then use QR decomposition again to calculate the orthogonal basis matrix of the matrix row vectors, obtaining the updated initialized matrix $Q$, Finally, obtain the initialized projection matrix $P$: $Q = QR(Q_{\text{prev}}, Q_{\text{init}})$, $P = I - Q^T Q$

The specific algorithm of our method is demonstrated in Appendix B

## 5 Experiments

### 5.1 Experiment set-up

**Datasets**. For the unlearn and retain task, we selected the widely recognized and popular ScienceQA dataset [25]. We collected pure text samples from this dataset to form a new dataset comprising 8,000 training samples and 2,000 test samples. Since the dataset primarily consists of three major subjects - *natural science*, *language science*, and *social science* - we chose four topics from *natural science* as continual unlearning requests: *physics*, *chemistry*, *biology*, and *earth science*. For the test samples, we selected the *language science* and *social science* subjects as the retain dataset to evaluate the model's commonsense reasoning capabilities. To verify the universality of our method across different data, we also conducted experiments using the WMDP dataset [21] as unlearning data. The specific experiments are detailed in Appendix D.1.

For robustness testing, we designed the following scenario: The attacker cannot obtain direct data related to the forgotten tasks but may access similar data for relearning attacks [16]. Specifically, we selected the *Genes to traits* category from the *biology* topic as the unlearn task, while using the *Classification* and *Heredity* categories from the same *biology* topic as similar data available to the attacker for relearning through fine-tuning.

**Metrics**. To evaluate the effectiveness of unlearning, we assessed model performance under the following conditions: For test tasks (all multiple-choice questions with varying options), we simulated complete unlearning by randomly guessing the answer and calculating the QA Accuracy as lower-bound accuracy $ACC_t^{low}$. The original model's performance served as the upper-bound accuracy $ACC_t^{up}$. The final score for task $t$ was calculated as: $S_t = (ACC_t - ACC_t^{low})/(ACC_t^{up} - ACC_t^{low})$ In continual unlearning experiments, after unlearning step $n$, we computed: (1)The average score across all unlearned tasks: $S_{unl} = \frac{1}{n}\sum_{i=1}^{n} S_{unl}^i$, (2)The average score across two retain tasks: $S_{ret} = \frac{1}{2}(S_{ret}^1 + S_{ret}^2)$. The final score was calculated as:$S_{ret} - S_{unl}$. This score captures both unlearning effectiveness and model utility preservation.

In addition to continuous unlearning scenarios, we evaluated all methods under a non-sequential unlearning setting. This simulates the case where model trainers have access to all unlearn data simultaneously and perform a single unlearning fine-tuning operation on the combined dataset. We measured the performance of this approach using score $Score_{all} = \frac{1}{2}(S_{ret}^1 + S_{ret}^2) - S_{unl}^{all}$, where $S_{unl}^{all}$ represents all the unlearn data.

For robustness testing, after task unlearning and relearning, we directly measure the QA Accuracy of the model on the unlearn and retain datasets as our evaluation metric.

**Compared Baselines**. To demonstrate our method's advantages, we selected several established baselines from machine unlearning literature, including both classical approaches and recent state-of-the-art methods. Specifically, we compared against GA [39], EUL [6], NPO [42], and RMU [21],O3 [13] with appropriate adaptations for the continual unsupervised learning scenario.

**Implementation Details**. Following Tofu [28] and O3 [13], we used LLaMA2-7B as our target model, Another popular model Qwen-7B [2] is also used in our experiment. All experiments are run repeatedly with three random cases, including random projected layers and random continual unlearning order, All experimental results were computed with the mean and 0.5× standard deviation to better evaluate the stability of different methods. We use the AdamW optimizer with 2e-4 as the learning rate and 5 as the batch size for both the unlearn and retain datasets. The default epochs are 2, and the LoRA rank for all experiments is 8. In each unlearning session, we select 200 unlearned and retain data to initialize the projection matrix. For each unlearning task, the dimension of the Q matrix in the projection matrix increases by 2. More details can be found in the Appendix C.

### 5.2 Overall Results

**Comparison with non-sequential unlearning** When comparing with $Score_{all}$ (the model performance after unified unlearning of all four tasks simultaneously), we find that most baselines show similar performance between $Score_1$ (single-task unlearning) and $Score_{all}$. This suggests that unified unlearning can indeed produce satisfactory results. However, their $Score_4$ scores (after four sequential unlearning operations) degrade dramatically, indicating these baseline methods fundamentally struggle with continual unlearning. Our method achieves an $Score_4$ score that differs by

| Method | $Score_1$ | $Score_2$ | $Score_3$ | $Score_4$ | $Score_{\text{all}}$ |
|---|---|---|---|---|---|
| GA | $0.585 \pm 0.078$ | $0.603 \pm 0.072$ | $0.529 \pm 0.063$ | $0.387 \pm 0.209$ | $0.675 \pm 0.086$ |
| EUL | $0.577 \pm 0.065$ | $0.428 \pm 0.061$ | $0.418 \pm 0.049$ | $0.208 \pm 0.075$ | $0.634 \pm 0.070$ |
| NPO | $\mathbf{0.982 \pm 0.075}$ | $0.327 \pm 0.035$ | $0.204 \pm 0.047$ | $0.172 \pm 0.038$ | $0.943 \pm 0.062$ |
| RMU | $0.777 \pm 0.101$ | $0.330 \pm 0.054$ | $0.108 \pm 0.060$ | $0.104 \pm 0.067$ | $0.767 \pm 0.040$ |
| O3 | $0.889 \pm 0.036$ | $0.784 \pm 0.026$ | $0.793 \pm 0.058$ | $0.785 \pm 0.031$ | $0.878 \pm 0.025$ |
| Ours | $0.950 \pm 0.022$ | $\mathbf{0.878 \pm 0.018}$ | $\mathbf{0.896 \pm 0.019}$ | $\mathbf{0.905 \pm 0.041}$ | $\mathbf{0.988 \pm 0.038}$ |

Table 2: Performance scores of different unlearning methods on llama2-7b

only 0.076 from the $Score_{\text{all}}$ benchmark (0.812 vs. 0.888), demonstrating its superior capability for continual knowledge removal while preserving model utility. In terms of method stability, our approach achieves an average standard deviation of 0.028, while most baselines exhibit higher average deviations (above 0.050). This demonstrates the superior stability of our method, as it consistently performs well across varying sequential unlearning tasks. Additionally, to demonstrate the generality of our method, we conducted the same experiments on the Qwen2.5-7B model and observed that our approach still outperforms other baselines, maintaining an unlearning performance score of 0.938 after 4 unlearn tasks, which surpasses the best baseline score of 0.762. Detailed experimental results can be found in Appendix D.2.

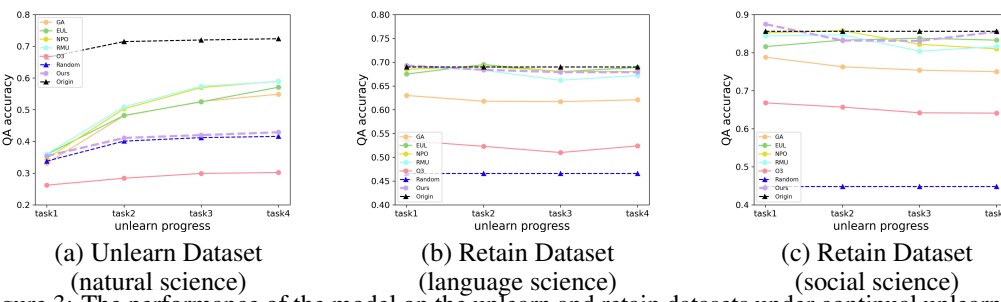

(a) Unlearn Dataset
(natural science)

(b) Retain Dataset
(language science)

(c) Retain Dataset
(social science)

Figure 3: The performance of the model on the unlearn and retain datasets under continual unlearning

Beyond overall scores, we conducted detailed analysis of model performance on unlearn tasks. As illustrated in Figure 3, *Origin* represents the original model accuracy, while *Random* denotes the accuracy when the model answers randomly. Our research shows that the O3 method exhibits significantly lower accuracy than *Random* (0.27 vs. 0.31), indicating detrimental effects on the model's fundamental capabilities (e.g., answering multiple-choice questions), which is an undesirable side effect that proper unlearning should avoid. Other four baseline methods gradually approach the *Origin* model's accuracy as unlearning progresses, suggesting incomplete unlearning. Our method maintains accuracy comparable to *Random* (0.30 vs. 0.31) even after four consecutive unlearning operations, demonstrating both immediate and persistent unlearning effectiveness.

By analyzing the response situation of the retain dataset, we can see that: GA and O3's severe performance degradation (respectively 10% and 20% drop) on retain tasks confirms its damaging impact on model capabilities. Other baseline methods preserve most retain accuracy, the performance difference compared to the original model remains within a 5% margin, showing their ability to maintain retain knowledge. These results collectively demonstrate that our approach uniquely satisfies both unlearning completeness and knowledge preservation requirements in continual learning scenarios.

In addition, we conduct identical experiments on the WMDP dataset. Our method achieved the highest score of 0.891 after continuous unlearning across three datasets, demonstrating significant improvement over the baseline's highest score of 0.810. These results confirm our method's broad effectiveness across different datasets. Detailed experimental results can be found in Appendix D.1.

## 5.3 Unlearning without same retain dataset

We also considered another scenario where keeping the retain dataset for an extended period may be impractical. In such cases, we can only perform the retain task using data similar to the retain dataset. To simulate this condition, we only keep one of the language science or social science datasets as

the training retain set, while using the other as the testing retain set. The experimental results are demonstrated in Table 3 and Table 4

Table 3: Performance score of the model when only using language science as the training retain dataset

| Method | $Score_1$ | $Score_2$ | $Score_3$ | $Score_4$ | $Score_{all}$ |
|---|---|---|---|---|---|
| GA | 0.440 ± 0.007 | 0.464 ± 0.107 | 0.204 ± 0.084 | 0.213 ± 0.063 | 0.415 ± 0.055 |
| EUL | 0.756 ± 0.028 | 0.273 ± 0.052 | 0.243 ± 0.102 | 0.133 ± 0.075 | 0.777 ± 0.047 |
| NPO | 0.829 ± 0.053 | 0.653 ± 0.030 | 0.474 ± 0.037 | 0.378 ± 0.173 | 0.834 ± 0.017 |
| RMU | 0.804 ± 0.086 | 0.484 ± 0.038 | 0.498 ± 0.034 | 0.415 ± 0.009 | 0.818 ± 0.069 |
| O3 | 0.764 ± 0.056 | 0.746 ± 0.040 | 0.654 ± 0.097 | 0.710 ± 0.045 | 0.804 ± 0.015 |
| Ours | **0.863 ± 0.029** | **0.836 ± 0.007** | **0.815 ± 0.023** | **0.836 ± 0.042** | **0.857 ± 0.028** |

Table 4: Performance score of the model when only using social science as the training retain dataset

| Method | $Score_1$ | $Score_2$ | $Score_3$ | $Score_4$ | $Score_{all}$ |
|---|---|---|---|---|---|
| GA | 0.592 ± 0.084 | 0.205 ± 0.073 | 0.113 ± 0.103 | 0.237 ± 0.091 | 0.452 ± 0.065 |
| EUL | 0.758 ± 0.081 | 0.134 ± 0.079 | 0.287 ± 0.118 | 0.072 ± 0.171 | 0.684 ± 0.064 |
| NPO | **0.843 ± 0.037** | 0.691 ± 0.083 | 0.448 ± 0.021 | 0.314 ± 0.099 | 0.789 ± 0.033 |
| RMU | 0.789 ± 0.067 | 0.435 ± 0.092 | 0.570 ± 0.094 | 0.504 ± 0.124 | **0.920 ± 0.081** |
| O3 | 0.603 ± 0.086 | 0.708 ± 0.049 | 0.661 ± 0.044 | 0.675 ± 0.116 | 0.802 ± 0.062 |
| Ours | 0.822 ± 0.043 | **0.826 ± 0.025** | **0.829 ± 0.018** | **0.792 ± 0.090** | 0.851 ± 0.040 |

Our findings demonstrate that our method remains effective even under these conditions, maintaining an average unlearning performance score of 0.814 after 4 unlearn tasks, which surpasses the highest average baseline score of 0.693. Moreover, our method exhibits a lower score standard deviation, indicating that its stability remains robust even in the absence of a retain dataset. Additional results about model performance on unlearn tasks and retain tasks are provided in Appendix D.3.

## 5.4 Robustness evaluation

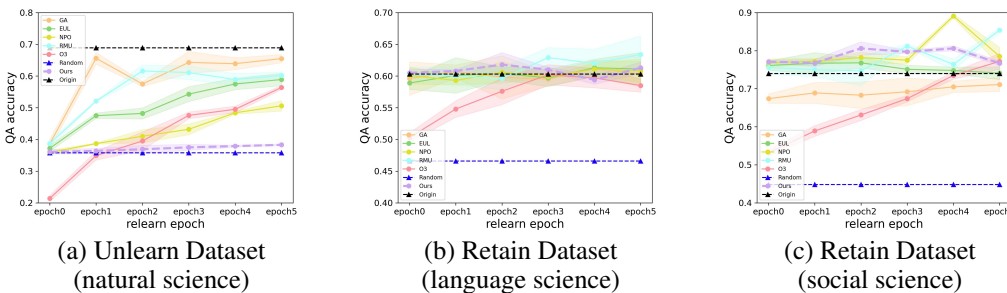

(a) Unlearn Dataset (natural science)

(b) Retain Dataset (language science)

(c) Retain Dataset (social science)

Figure 4: The performance of the model on the unlearn and retain datasets under relearn attack

As shown in Figure 4, our method also demonstrates promising effectiveness against relearn attacks. While most methods achieve satisfactory performance in the unlearn task, successfully reducing the accuracy on the unlearn dataset below 0.4 (close to the random guess lower bound of 0.35), they show vulnerability when facing relearn attacks. Notably, even when the relearn dataset is similar to yet distinct from the unlearn dataset, most methods exhibit accuracy exceeding 0.45 after just the first relearn epoch. After 5 epochs, these methods typically reach an accuracy around 0.5. This observation strongly suggests that these approaches do not genuinely erase knowledge but rather mask the model's activation patterns for specific knowledge through parameter fine-tuning. In contrast, our method maintains accuracy below 0.4 even after 5 relearn epochs, demonstrating superior robustness in ensuring complete unlearning of the unlearn dataset knowledge.

For the retain dataset, our experiments reveal that the relearning also improves the performance of the model on the retain dataset. This phenomenon suggests that the fine-tuning process on the relearn dataset may provide beneficial transfer effects to the retain dataset, potentially enhancing its performance through shared feature representations. Moreover, our method demonstrates a narrower

0.5× standard deviation range in accuracy for both unlearn and retain datasets, further evidencing its superior stability.

## 5.5 Computational cost comparison

Our method exhibits advantages in both parameter efficiency and computational speed, as quantified in Table 5. During continuous unlearning of four tasks, the approach achieves superior performance with merely 0.1M trainable parameters and gain an order of magnitude

| method | GA | EUL | NPO | RMU | O3 | Ours |
|---|---|---|---|---|---|---|
| #Trainable Parameters | 4.2M | 33.7M | 4.2M | 4.2M | 20M | **0.1M** |
| Clock time per batch (s) | 1.02 | 1.28 | 0.73 | 0.89 | 1.64 | **0.71** |

Table 5: computational cost comparison across different methods

reduction, compared to the 4.2M parameters required by conventional LoRA-based methods. Furthermore, runtime measurements reveal our method's computational superiority, processing batches in 0.71 seconds (vs. 0.89-1.28s for alternatives), representing a 20-45% speedup. These combined efficiencies make the approach particularly suitable for resource-constrained environments.

## 5.6 Ablation Study

**Removing matrix initialization**. In our framework, the initialization of projection matrices constitutes a crucial component. To validate its effectiveness, we conduct an ablation study by removing this initialization step. When the matrix is randomly initialized, experimental results demonstrate that the training requires 5 epochs to achieve optimal performance (with other hyperparameters kept at default settings).

| Method | $Score_1$ | $Score_2$ | $Score_3$ | $Score_4$ | $Score_{all}$ |
|---|---|---|---|---|---|
| w/ initial | **0.950 ± 0.022** | **0.878 ± 0.018** | **0.896 ± 0.019** | **0.905 ± 0.041** | **0.988 ± 0.038** |
| w/o initial | 0.881 ± 0.054 | 0.761 ± 0.057 | 0.768 ± 0.045 | 0.655 ± 0.072 | 0.856 ± 0.056 |

Table 6: Performance comparison of our method with (w/) or without (w/o) the projection matrix initialization

As evidenced in Table 6, the non-initialized projection matrix exhibits significantly inferior performance. The initial score begins below 0.9, and further deteriorates to 0.65 after unlearning four consecutive tasks. This suggests that merely fine-tuning randomly initialized parameters still adversely affects the model's retention capability. In contrast, our orthogonal initialization method provides two key advantages: (1) it preserves model performance on retain tasks by enforcing orthogonality between the projection directions and retain data, and (2) it substantially reduces computational resources - achieving competitive results within just 2 training epochs compared to the 5 epochs required by the random initialization baseline.

In our experiments, the number of projection layers and the dimensionality of projection matrices emerged as two critical hyperparameters. Through comprehensive hyperparameter studies, we observed that the model maintains a robust unlearning score of 0.890 after 4 unlearn tasks even when reduced to just 1 projection layer. Notably, increasing the projection matrix dimensionality from 1 to 2 yields a dramatic performance leap, improving the unlearning score from 0.591 to 0.905. Detailed results can be found in Appendix D.4.

## 6 Conclusion

In this work, we presented Metamorphosis Representation Projection (MRP), a novel method for effective continual unlearning in large language models. Our work provides a comprehensive solution to two persistent challenges in machine unlearning—continuous unlearning and defense against relearning attacks. Through both theoretical analysis and extensive experiments, we demonstrate that the proposed method: (1) achieves stable performance in sequential unlearning scenarios without catastrophic forgetting, and (2) effectively prevents knowledge recovery via relearning attacks. These contributions advance the field by establishing a new paradigm for representation-level unlearning while offering practical insights for real-world deployment. In addition, we discussed the content of the article on limitations and future work in Appendix E.

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

# Appendix

## A  Proof of lemma 3.1 and lemma 3.2

### A.1  Definitions

**Definition A.1.** *A matrix $P \in \mathbb{R}^{n \times n}$ is called a* projection matrix *if it is idempotent, i.e., $P^2 = P$.*

**Definition A.2.** *A matrix $Q \in \mathbb{R}^{k \times n}$ is called* orthogonal *if its rows are orthonormal vectors, i.e., $QQ^T = I_k$ where $I_k$ is the $k \times k$ identity matrix.*

### A.2  Proof of Spectral Theorem

Before formally proving lemma 3.1, we first need to prove a fundamental theorem.

**theorem A.1** (Spectral Theorem for Real Symmetric Matrices). *Let $A \in \mathbb{R}^{n \times n}$ be a real symmetric matrix ($A = A^T$, all eigenvalues of A are real). Then A can be diagonalized as:*

$$A = Q\Lambda Q^T$$

*where $\Lambda = diag(\lambda_1, \ldots, \lambda_n)$ contains the eigenvalues of A, and Q is an orthogonal matrix ($Q^T Q = I$) whose columns are the corresponding eigenvectors.*

*Proof.* We proceed by induction on the dimension $n$.

**Base case ($n = 1$):**  Any $1 \times 1$ matrix is trivially diagonal, and the theorem holds.

**Inductive step:**  Assume the theorem holds for all symmetric matrices of size $(n-1) \times (n-1)$.

**Step 1: Existence of real eigenvalues.** Consider $A$ as a linear operator on $\mathbb{C}^n$. The characteristic polynomial $p_A(\lambda) = \det(A - \lambda I)$ has at least one complex root $\lambda_1$ by the Fundamental Theorem of Algebra. Let $\mathbf{q_1} \in \mathbb{C}^n$ be a corresponding eigenvector ($(A - \lambda_1 I)\mathbf{q_1} = 0$) with $\|\mathbf{q_1}\| = 1$.

Since $A$ is real and symmetric:

$$\lambda_1 = \mathbf{q_1}^* \lambda_1 I \mathbf{q_1} = \mathbf{q_1}^* A^T \mathbf{q_1} = (A\mathbf{q_1})^* \mathbf{q_1} = \overline{\lambda_1} \mathbf{q_1}^* \mathbf{q_1} = \overline{\lambda_1}$$

where $\mathbf{q_1}^*$ is the conjugate transpose of $\mathbf{q_1}$
Thus $\lambda_1 \in \mathbb{R}$.

**Step 2: Construction of orthonormal basis.** Let $\lambda_1$ be an eigenvalue of $A$ with corresponding unit eigenvector $\mathbf{q}_1 \in \mathbb{R}^n$. Extend $\mathbf{q}_1$ to an orthonormal basis $\{\mathbf{q}_1, \mathbf{w}_2, \ldots, \mathbf{w}_n\}$ of $\mathbb{R}^n$.

Let $Q_1 = [\mathbf{q}_1 \ \mathbf{w}_2 \ \cdots \ \mathbf{w}_n]$. Then:

$$Q_1^T A Q_1 = \begin{bmatrix} \lambda_1 & \mathbf{0}^T \\ \mathbf{0} & A_{n-1} \end{bmatrix}$$

where $A_{n-1}$ is $(n-1) \times (n-1)$ and symmetric (since $(Q_1^T A Q_1)^T = Q_1^T A^T Q_1 = Q_1^T A Q_1$).

**Step 3: Induction.** By the induction hypothesis, $A_{n-1}$ has an orthonormal eigenbasis $\{\mathbf{q}_2, \ldots, \mathbf{q}_n\}$. Let:

$$Q_2 = \begin{bmatrix} 1 & \mathbf{0}^T \\ \mathbf{0} & \widetilde{Q} \end{bmatrix}, \quad \text{where } \widetilde{Q} = [\mathbf{q}_2 \cdots \mathbf{q}_n]$$

Then $Q = Q_1 Q_2$ is orthogonal and:

$$Q^T A Q = \Lambda = \text{diag}(\lambda_1, \ldots, \lambda_n)$$

Thus $A = Q\Lambda Q^T$.

**Step 4: Orthogonality of eigenvectors.** For distinct eigenvalues $\lambda_i \neq \lambda_j$:

$$\lambda_i \mathbf{q}_i^T \mathbf{q}_j = (A\mathbf{q}_i)^T \mathbf{q}_j = \mathbf{q}_i^T A^T \mathbf{q}_j = \mathbf{q}_i^T A \mathbf{q}_j = \lambda_j \mathbf{q}_i^T \mathbf{q}_j$$

Thus $(\lambda_i - \lambda_j)\mathbf{q}_i^T \mathbf{q}_j = 0 \implies \mathbf{q}_i^T \mathbf{q}_j = 0$.

For repeated eigenvalues, we can choose orthonormal eigenvectors via Gram-Schmidt. $\qquad \square$

**corollary A.1.1** (Spectral Decomposition). *Any real symmetric matrix $A$ can be written as:*

$$A = \sum_{i=1}^{n} \lambda_i \mathbf{q}_i \mathbf{q}_i^T$$

*where $\lambda_i$ are eigenvalues and $\{\mathbf{q}_i\}$ form an orthonormal basis.*

### A.3 Proof of lemma 3.1

*Proof of Lemma 3.1.* Let $P$ be a projection matrix. Since $P$ is idempotent, it can be diagonalized with eigenvalues either 0 or 1.

Let $k = \text{rank}(I - P)$. By theorem A.1, There exists an orthogonal matrix $U \in \mathbb{R}^{n \times n}$ such that:

$$P = U \begin{bmatrix} I_{n-k} & 0 \\ 0 & 0 \end{bmatrix} U^T$$

where $I_{n-k}$ is the $(n - k) \times (n - k)$ identity matrix.

Then:

$$I - P = U \begin{bmatrix} 0 & 0 \\ 0 & I_k \end{bmatrix} U^T$$

Let $Q \in \mathbb{R}^{k \times n}$ consist of the last $k$ rows of $U^T$. Since $U$ is orthogonal, the rows of $Q$ are orthonormal, making $Q$ an orthogonal matrix. We can then write:

$$I - P = \sum_{i=n-k+1}^{n} u_i u_i^T = Q^T Q$$

where $u_i$ are the columns of $U$.

Therefore, we have shown that $P = I - Q^T Q$ for some orthogonal matrix $Q$. $\qquad\square$

### A.4 Proof of lemma 3.2

*Proof of Lemma 3.2.* Given $P = I - Q^T Q$ where $Q$ is orthogonal, we need to show that for any $x \in \mathbb{R}^n$, $Px$ is the orthogonal projection onto the complement of the row space of $Q$.

Let $\mathcal{R}(Q)$ denote the row space of $Q$ and $\mathcal{R}(Q)^{\perp}$ its orthogonal complement.

1. **Projection property**: First, verify that $P$ is indeed a projection:

$$\begin{aligned}
P^2 &= (I - Q^T Q)(I - Q^T Q) \\
&= I - 2Q^T Q + Q^T Q Q^T Q \\
&= I - Q^T Q \quad (\text{since} QQ^T = I_k) \\
&= P
\end{aligned}$$

2. **Range space**: For any $x \in \mathbb{R}^n$, $Px = x - Q^T Qx$. Note that:

$$QPx = Q(x - Q^T Qx) = Qx - (QQ^T)Qx = Qx - Qx = 0$$

so $x - xQ^T Q$ is orthogonal to $\mathcal{R}(Q)$ Therefore, $xP$ is indeed the orthogonal projection of $x$ onto the complement of the row space of $Q$. $\qquad\square$

## B  Algorithm of MRP

Our algorithm is presented as follows, with detailed procedures available in Section Proposed Method 4 of the paper

**Algorithm 1** Continual Unlearning algorithm

**Input**: Unlearn datasets $D_{u1} \sim D_{un}$, Retain dataset $D_r$, hyperparameter $\alpha$, Learning rate $\beta$, Projection layer set $L$, Origin Model $f^W$, Model Parameter $W$, Projection dimension $k$
**Output**: Unlearned model $f^{W,P}$

1: **for** $i \in [1...n]$ **do**
2:    **for** $l \in L$ **do**
3:       Record previous projection matrices $P_{l,prev} = I - Q_{l,prev}^T Q_{l,prev}$
4:       compute Hidden state principal components matrix of retain tasks $H_{l,r}$
5:       Using QR decomposition to calculate the orthogonal basis of $H_{l,r}$: $Q_{l,r} = QR(H_{l,r}^T)$
6:       compute Hidden state principal components matrix of unlearn tasks $H_{l,ui}$
7:       Calculate the principal components after projecting $H_{l,ui}$: $H_{l,i} = PCA((I - Q_{l,r}^T Q_{l,r})H_{l,ui}^T)$
8:       Combine $Q_{l,prev}$ and $H_{l,i}$ and calculate their orthogonal basis $Q_{l,i} = QR(Q_{l,prev}^T, H_{l,i}^T)$
9:       Update projection matrix $P_{l,i} = I - Q_{l,i}^T Q_{l,i}$
10:      $P_i = \{P_{l,i} | l \in L\}, Q_i = \{Q_{l,i} | l \in L\}$
11:      Use the hook function to convert $f^W$ to $f^{W,P_i}$
12:    **end for**
13:    $\mathcal{L}(W, Q_i, d_u, d_r) = \alpha L(W, P_i, d_r) - L(W, P_i, d_u)$
14:    **for** $(d_u, d_r)$ in $(D_{ui}, D_r)$ **do**
15:       **for** $l$ in $L$ **do**
16:          $Q_{l,i} = Q_{l,i} - \beta \nabla_{Q_{l,i}} \mathcal{L}(W, Q_i, d_u, d_r)$
17:       **end for**
18:    **end for**
19:    **for** $l$ in $L$ **do**
20:       Update the previous projection matrix $Q_{l,prev} = Q_{l,i}$
21:    **end for**
22: **end for**
23: **return** Unlearned model $f^{W,P}$

## C   More Details on Experiments

### C.1   System prompts

We follow [34] to use a system prompt in the following box to build a supervised data set for fine-tuning.

> Below is an instruction that describes a task, paired with an input that provides further context. Write a response that appropriately completes the request. Instruction:**instruction** Input:**input** Response:**response**

For our experiment, we construct the following triplet of Instruction/Input/Response:

> **The triplet of Instruction/Input/Response for ScienceQA dataset:**
> Instruction: Choose the correct answer's letter,just like (A), (B), (C) or (D).
> Input: <Corresponding input in ScienceQA dataset>(use (A), (B), (C), (D) as option labels)
> Response: <Corresponding label in ScienceQA dataset>

> **The triplet of Instruction/Input/Response for WMDP dataset:**
> Instruction: Choose the correct answer's letter,just like (A), (B), (C) or (D).
> Input: <Corresponding input in WMDP dataset>(use (A), (B), (C), (D) as option labels)
> Response: <Corresponding label in WMDP dataset>

 **C.2   More details for baselines**

**GA**: We follow the hyperparameter settings in the paper  [39], set the learning rate of unlearn data to 0.1 and the learning rate of retain data to 1.
**EUL**: We follow the method described in the paper [6], plugging the unlearning layers into transformer layers after the feed-forward networks and using Low-Rank Adaptation to train the unlearning layers
**NPO**: We follow the hyperparameter settings in paper  [42] and selected the hyperparameter $\beta$ within the range of $0.01 \sim 1$. We found that $\beta = 0.2$ performed better, so we chose $\beta = 0.2$ as the experimental hyperparameter in the article
**RMU**: We follow the hyperparameter settings in paper [21] and select the hyperparameter $\alpha$ within the range of $10^{-4} \sim 10$. We found that $\alpha = 1$ yields better performance, and thus we choose $\alpha = 1$ as the experimental hyperparameter in our paper.
**O3**: We followed the method described in the paper [13], employing O-LoRA for model fine-tuning in continual unlearning, and training an OOD classifier for the ScienceQA dataset to make judgments.

# D   More Experiment Results

## D.1   Performance of MRP on WMDP dataset

### D.1.1   Set up

In our WMDP dataset, we selected a total of 2,000 samples as the unlearning training set and 800 samples as the test set for the unlearning task. For the unlearning task specifically, we chose three subjects from the WMDP dataset - biology, chemistry, and cybersecurity - as the data for continuous unlearning. The experiments were conducted with three randomized sequences, and the reported results represent the mean values ± 0.5 standard deviations.

Regarding the retain dataset, since WMDP doesn't contain appropriate retain data, we continued using the language science and social science subjects from Science QA as retain data. All other experimental parameters remained identical to those used in the Science QA dataset experiments.

### D.1.2   Result

Table 7: Performance scores of different unlearn methods on WMDP dataset

| Method | $Score_1$ | $Score_2$ | $Score_3$ | $Score_{\text{all}}$ |
|---|---|---|---|---|
| GA | 0.663 ± 0.033 | 0.767 ± 0.040 | 0.650 ± 0.063 | 0.843 ± 0.069 |
| EUL | 0.849 ± 0.062 | 0.521 ± 0.090 | 0.297 ± 0.022 | 0.790 ± 0.033 |
| NPO | 0.838 ± 0.053 | 0.386 ± 0.050 | 0.265 ± 0.114 | 0.895 ± 0.077 |
| RMU | 0.892 ± 0.088 | 0.406 ± 0.093 | 0.264 ± 0.078 | 0.872 ± 0.036 |
| O3 | 0.779 ± 0.022 | 0.798 ± 0.051 | 0.810 ± 0.035 | 0.817 ± 0.055 |
| Ours | **0.905 ± 0.020** | **0.872 ± 0.022** | **0.891 ± 0.047** | **0.902 ± 0.016** |

As shown in Table 7, our method maintains relatively high scores throughout the continuous unlearning process. Notably, after three consecutive unlearning operations, most baseline methods drop to scores of 0.65 or below, while our method consistently remains above 0.87. After three unlearning iterations, the difference between our method and the $Score_{\text{all}}$ is merely 0.011 points (0.891 vs. 0.902). These results demonstrate the broad effectiveness of our approach across the dataset.

## D.2   Performance of MRP on Qwen model

As shown in Table 8, most baseline methods maintain relatively high scores when unlearning the first task. However, their performance drops significantly to 0.8 or below during the second unlearning operation. After unlearning four tasks, the scores of most methods decline to 0.57 or lower. In contrast, our method consistently maintains a score above 0.9 throughout the continuous unlearning

Table 8: Performance scores of different unlearn methods on qwen7b

| Method | $Score_1$ | $Score_2$ | $Score_3$ | $Score_4$ | $Score_{all}$ |
|--------|-----------|-----------|-----------|-----------|---------------|
| GA  | 0.849 ± 0.018 | 0.436 ± 0.019 | 0.291 ± 0.037 | 0.271 ± 0.094 | 0.831 ± 0.010 |
| EUL | 0.916 ± 0.008 | 0.837 ± 0.010 | 0.654 ± 0.018 | 0.569 ± 0.038 | 0.925 ± 0.006 |
| NPO | 0.946 ± 0.016 | 0.805 ± 0.007 | 0.395 ± 0.015 | 0.407 ± 0.007 | 0.921 ± 0.006 |
| RMU | 0.945 ± 0.008 | 0.739 ± 0.016 | 0.326 ± 0.043 | 0.429 ± 0.019 | 0.953 ± 0.037 |
| O3  | 0.684 ± 0.014 | 0.783 ± 0.025 | 0.716 ± 0.018 | 0.762 ± 0.013 | 0.829 ± 0.026 |
| Ours | **0.964 ± 0.003** | **0.940 ± 0.011** | **0.926 ± 0.003** | **0.939 ± 0.012** | **0.975 ± 0.014** |

process. Another approach, O3, demonstrates reasonable performance in the sequential unlearning scenario, but our method achieves superior results across all evaluation metrics.

When also comparing with $Score_{all}$ (model performance after simultaneously unlearning all four tasks), we observe that most baselines exhibit similar performance between $Score_1$ (single-task unlearning) and $Score_{all}$, suggesting that unified unlearning can indeed yield satisfactory results. However, their $Score_4$ scores (after four sequential unlearning operations) drop sharply, indicating that these baseline methods fundamentally struggle with sustained unlearning.

In contrast, our method achieves a $Score_4$ that differs from the $Score_{all}$ benchmark by only 0.037 (0.938 vs. 0.975), demonstrating its superior capability in maintaining model utility while performing continual knowledge removal.

This result indicates that our method is still effective in different models, proving the universality of our method.

### D.3 Unlearning without same retain dataset

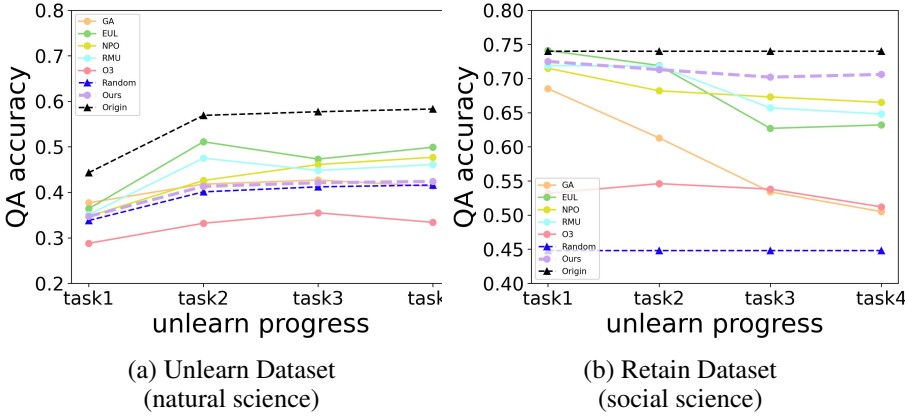

(a) Unlearn Dataset
(natural science)

(b) Retain Dataset
(social science)

Figure 5: The performance of the model on the unlearn and retain datasets when only use language science dataset as the training retain dataset

In Tables 3 and 4, we show overall scores of this experiment, beyond overall scores, we conducted detailed analysis of model performance on unlearn tasks and retain tasks.As demonstrated in Figure 5 and Figure 6, our method consistently maintains a lower QA accuracy on the unlearn tasks across both experiments. Even after four epochs of unlearn requests, the QA accuracy remains around 0.4, while most other methods exhibit an increase to approximately 0.5. Notably, although O3 achieves even lower accuracy on the unlearn task, its performance falls below that of Random, indicating that O3's approach compromises the model's capability in handling multiple-choice questions. This observation is further corroborated by the performance on the retain datasets.

Our method demonstrates robust performance on the retain datasets in both experiments. On the social science dataset, the accuracy drops by less than 0.05 compared to the original model, whereas

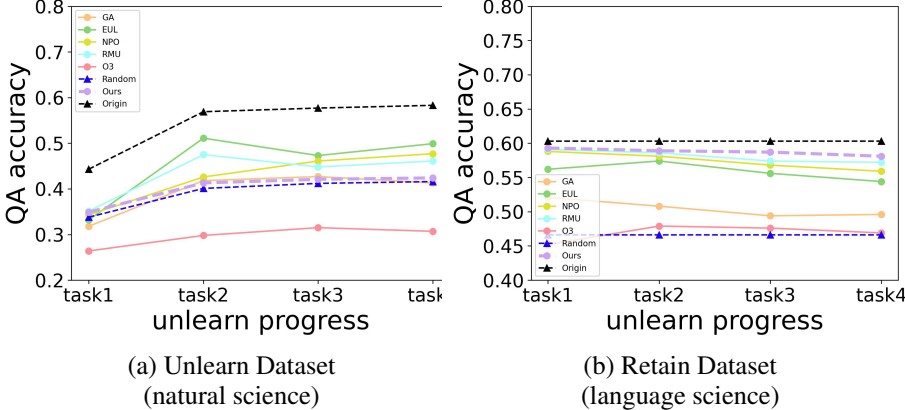

(a) Unlearn Dataset
(natural science)

(b) Retain Dataset
(language science)

Figure 6: The performance of the model on the unlearn and retain datasets when only use social science dataset as the training retain dataset

most other methods suffer a decline of around 0.15. Particularly, O3 experiences a significant drop of 0.20, highlighting its detrimental impact on the model's performance in other tasks.

These results strongly suggest that, even in the absence of the original retain dataset, our approach can effectively utilize similar datasets as substitutes to achieve competitive retain performance.

### D.4 hyperparameter experiments

In this section, we conducted hyperparameter experiments on two critical parameters: the projection dimension and the number of projection layers. For the projection dimension, we tested values from 1 to 4 while maintaining a fixed learning rate of $2 \times 10^{-4}$ and 2 projection layers. For the projection layers, we also tested layer numbers from 1 to 4 while maintaining a fixed learning rate of $2 \times 10^{-4}$ and 2 projection layers.Other hyperparameters kept at default settings.

Table 9: Performance scores of different projection dimensions

| Dim | $Score_1$ | $Score_2$ | $Score_3$ | $Score_4$ | $Score_{all}$ |
|---|---|---|---|---|---|
| 1 | $0.445 \pm 0.053$ | $0.669 \pm 0.049$ | $0.603 \pm 0.080$ | $0.591 \pm 0.061$ | $0.454 \pm 0.057$ |
| 2 | $0.950 \pm 0.021$ | $\mathbf{0.878 \pm 0.027}$ | $\mathbf{0.896 \pm 0.027}$ | $\mathbf{0.905 \pm 0.079}$ | $0.988 \pm 0.074$ |
| 3 | $1.113 \pm 0.026$ | $0.876 \pm 0.012$ | $0.863 \pm 0.033$ | $0.901 \pm 0.082$ | $0.982 \pm 0.033$ |
| 4 | $\mathbf{1.223 \pm 0.017}$ | $0.853 \pm 0.014$ | $0.855 \pm 0.031$ | $0.884 \pm 0.021$ | $\mathbf{1.016 \pm 0.035}$ |

Table 10: Performance scores of different number of projection layers

| Num | $Score_1$ | $Score_2$ | $Score_3$ | $Score_4$ | $Score_{all}$ |
|---|---|---|---|---|---|
| 1 | $0.480 \pm 0.084$ | $0.863 \pm 0.016$ | $0.838 \pm 0.065$ | $0.890 \pm 0.128$ | $0.897 \pm 0.059$ |
| 2 | $\mathbf{0.950 \pm 0.019}$ | $\mathbf{0.878 \pm 0.018}$ | $\mathbf{0.896 \pm 0.024}$ | $\mathbf{0.905 \pm 0.046}$ | $\mathbf{0.988 \pm 0.103}$ |
| 3 | $0.668 \pm 0.025$ | $0.804 \pm 0.008$ | $0.859 \pm 0.004$ | $0.860 \pm 0.065$ | $0.962 \pm 0.068$ |
| 4 | $0.755 \pm 0.020$ | $0.835 \pm 0.019$ | $0.881 \pm 0.023$ | $0.855 \pm 0.010$ | $0.944 \pm 0.021$ |

### D.4.1 Projection Dimension Analysis

As shown in Table 9, when the dimension equals 1, the model achieves relatively low performance (score ≈ 0.5), indicating insufficient capacity for effective unlearning. Higher dimensions yield significantly better results (scores ≈ 0.9), demonstrating improved unlearning effectiveness. Notably, the orthogonal constraint between projection directions and retain data preserves the retain dataset accuracy despite increasing dimensions.

### D.4.2 Projection Layer Analysis

Table 10 reveals optimal performance at 2 projection layers (final score: 0.905). Both insufficient and excessive layers degrade performance - the former limits unlearning effectiveness while the latter may interfere with retain dataset performance. This suggests careful layer count selection is crucial.

# E   Limitations and Future Work

Although our method performs robustly across a range of projection layer selections, specific layers can still be chosen for optimal results depending on the target model. Additionally, since this is the first application of projection-based methods to LLM unlearning, we have only validated MRP on smaller 7B language models. Its scalability to larger models or other types of models beyond LLMs (such as multimodal models) remains to be thoroughly investigated. Furthermore, the concept of projection can be widely applied to other safety-related domains, such as eliminating discriminatory or harmful information. Research in this direction will be a focus of our future work.

