# OpenReview forum: "Reliable Unlearning Harmful Information in LLMs with Metamorphosis Representation Projection"
_NeurIPS.cc/2025/Workshop/Reliable_ML — NeurIPS 2025 - Reliable ML Workshop_

### Official Review · Reviewer_DPSL · 2025-09-14
**Good contribution, presentation could be clearer**

**Rating:** 7
**Confidence:** 3

**Review:**

*Summary*

This paper discusses new technique in doing machine unlearning in the field of LLM, defined as retraining a machine learning model that "forgets" a part of the dataset that it has already been trained on. The paper first identifies the limitations of existing literature in machine unlearning in LLM (catastrophic forgetting and do not show complete elimination of the model knowledge), then outlines their methodology in addressing them. In particular, it exhibits the presence of catastrophic forgetting in sequential machine unlearning of the existing methods, by showing that the gradients fluctuate in signs when moving to subsequent unlearning tasks (Section 3.1). The second part of the contribution of this paper is a new method (MRP) to unlearn by using parameter projection by aligning the decomposed matrix (Q) in the directions of unlearned task, then using $$P=I-Q^TQ$$ as the parameter projection matrix. The paper then validates this claim on the ScienceQA dataset, using natural science as train (unlearning) dataset and social and language sciences as test (retaining) dataset, and uses the difference of retaining and unlearning accuracy as the metric. Overall, the paper shows that the new method outperforms the existing methods. The paper also shows robustness of their methods against "relearning" attacks, i.e. another improvement to the existing frameworks. In the ablation study the paper shows that matrix initialization is essential.

*Strengths*

Here are some of the reasons why I think this paper is worth accepting:

- It makes an interesting contribution on machine unlearning in LLM, including tackling the problem of sequential unlearning.

- The experiment procedures seem to be well-executed: it shows that their MRP method can effectively retain the existing knowledge on the retain set while erasing the knoweldge on unlearn set. It also shows robustness against unlearn set which shows that the models are not only superficially surpressing the "correct" outputs like the baselines.

*Weaknesses*

The main weaknesses typically come in making the paper clearer, e.g.:

- Line 153: "Let Then"

- Lines 158, 160: "lemma" (capitalize)

- A short definition of catastrophic unlearning would have been nice.

In addition to these, I thought that the paper would have been more interesting if it can demonstrate some experiments on the actual use cases of machine unlearning (e.g. sensitive PII or harmful data).

Question: in Lemmas 3.1 and 3.2, what does the dimension $$k$$ imply (is that the number of unlearning topics)?

---

### Official Review · Reviewer_RsKt · 2025-09-19

**Rating:** 7
**Confidence:** 3

**Review:**

Summary

The authors propose Metamorphosis Representation Projection (MRP) as a method for reliable unlearning in large language models. The idea is to apply projection operations in hidden state space to remove harmful information while preserving useful knowledge. Experiments on ScienceQA, WMDP, and two 7B models show MRP achieves better continual unlearning performance, is more robust against relearning attacks, and is more efficient compared to baseline methods.

Strengths

1. This work tackles an important problem in LLM safety, particularly the challenge of continual unlearning where existing methods often fail.
2. The projection-based approach is novel and the empirical evaluation is extensive, covering multiple datasets and model backbones.
3. The efficiency results are compelling, since the method requires significantly fewer trainable parameters and less compute.

Weaknesses / Limitations

1. The projection approach is mathematically sound but the explanation is not intuitive, making accessibility limited.
2. The method is only tested on medium-scale open models and QA-style datasets, leaving questions about scalability to larger or more diverse LLMs.
3. The choice of projection initialization is critical yet not fully justified. Finally, the paper claims irreversibility of forgetting, but this is only demonstrated empirically without stronger theoretical guarantees.

Suggestions for Authors

1. Evaluate on real-world unlearning cases such as PII or jailbreak datasets to strengthen practical relevance.
2. Provide a clearer explanation of the projection intuition for non-experts.
3. Include additional ablations on initialization and overlap between unlearn and retain data.
4. Discuss more explicitly how this approach could be integrated into regulatory compliance workflows.

Ethics
- The motivation is well-grounded in privacy and “right to be forgotten” regulations. Still, unlearning can have fairness implications if harmful knowledge is entangled with subgroup knowledge. There is also a dual-use risk where projection could selectively remove alignment safeguards. These should be acknowledged along with mitigation strategies.